# Peroxisome Proliferator-Activated Receptor Beta/Delta Agonist Suppresses Inflammation and Promotes Neovascularization

**DOI:** 10.3390/ijms21155296

**Published:** 2020-07-26

**Authors:** Yutaro Tobita, Takeshi Arima, Yuji Nakano, Masaaki Uchiyama, Akira Shimizu, Hiroshi Takahashi

**Affiliations:** 1Department of Ophthalmology, Nippon Medical School, Bunkyo-ku, Tokyo 113-8603, Japan; y-tobita@nms.ac.jp (Y.T.); n-yuji@nms.ac.jp (Y.N.); uchiyama@nms.ac.jp (M.U.); tash@nms.ac.jp (H.T.); 2Department of Analytic Human Pathology, Nippon Medical School, Bunkyo-ku, Tokyo 113-8603, Japan; ashimizu@nms.ac.jp

**Keywords:** corneal inflammation, corneal neovascularization, PPARβ/δ, alkali burn

## Abstract

The effects of peroxisome proliferator-activated receptor (PPAR)β/δ ophthalmic solution were investigated in a rat corneal alkali burn model. After alkali injury, GW501516 (PPARβ/δ agonist) or vehicle ophthalmic solution was topically instilled onto the rat’s cornea twice a day until day 7. Pathological findings were evaluated, and real-time reverse transcription polymerase chain reaction was performed. GW501516 strongly suppressed infiltration of neutrophils and pan-macrophages, and reduced the mRNA expression of interleukin-6, interleukin-1β, tumor necrosis factor alpha, and nuclear factor-kappa B. On the other hand, GW501516 promoted infiltration of M2 macrophages, infiltration of vascular endothelial cells associated with neovascularization in the wounded area, and expression of vascular endothelial growth factor A mRNA. However, 7-day administration of GW501516 did not promote neovascularization in uninjured normal corneas. Thus, the PPARβ/δ ligand suppressed inflammation and promoted neovascularization in the corneal wound healing process. These results will help to elucidate the role of PPARβ/δ in the field of ophthalmology.

## 1. Introduction

Peroxisome proliferator-activated receptor (PPAR) is a nuclear receptor [1,2]. PPAR has three subtypes, α, β (δ), and γ, each of which is activated by a specific ligand and controls expression of various genes by binding to the regulatory region of the target gene [1]. PPAR has been investigated as a target for the treatment of lipid metabolism, hyperlipidemia, diabetes, and atherosclerosis [2,3,4]. In clinical practice, fenofibrate, a PPARα ligand, is used as a therapeutic agent for hyperlipidemia, and pioglitazone, a PPARγ ligand, is used for diabetes [2,5]. However, PPARβ/δ ligands have not yet been used as therapeutic drugs for any diseases [2]. PPARβ/δ is expressed in many tissues and organs such as skeletal muscle, adipose tissue, the cardiovascular system, the uterus after implantation, intestine, brain, and skin, and is involved in many important functions, such as energy metabolism, cell differentiation and proliferation, tissue repair, and cancer progression, suggesting its potential as a therapeutic target [2,6,7].

In addition to these basic functions, PPARs are also anti-inflammatory [8]. Therefore, we previously investigated the clinical application of PPARs in the field of ophthalmology and demonstrated anti-inflammatory and anti-angiogenic effects of PPARα [9,10] and γ [11] ligands using a rat corneal alkali burn model. Like other subtypes, PPARβ/δ is considered to play roles in corneal wound healing. In this study, we investigated the characteristics of GW501516, a PPARβ/δ ligand, and explored the therapeutic effects of an ophthalmic solution of a PPARβ/δ agonist using a rat alkali burn model.

## 2. Results

### 2.1. Wound Healing Process and Expression of PPARβ/δ

Hematoxylin-eosin (HE) staining was conducted to compare the corneal wound healing process between the vehicle group and the PPARβ/δ agonist (GW501516) instilled group (Figure 1a–p). The corneal epithelial layer disappeared in the area in which alkali-immersed paper was placed. In the early phase after injury, stromal edema was observed in the center of the cornea. We noted regeneration of the corneal epithelium on day 1. Infiltration of inflammatory cells was observed in the corneal limbus from the early phase after injury. On day 4, infiltration of inflammatory cells moved to the central cornea. On day 4 and day 7, luminal structures associated with neovascularization increased in the corneal limbus. Less infiltration of inflammatory cells was observed (Figure 1q) and more neovascularization was seen in the PPARβ/δ group compared to the vehicle group (Figure 1d,h,l,p). In the macroscopic images, corneal transparency was better in the PPARβ/δ group on day 3 (Figure 1r,s). However, on day 7, corneal neovascularization and anterior chamber bleeding were more obvious in the PPARβ/δ group (Figure 1t,u). It was suggested that GW501516 suppressed inflammation, and improved corneal transparency in the early stage of the injury, but promoted neovascularization, increased anterior chamber bleeding, and eventually, impaired intraocular transparency. We investigated expression of PPARβ/δ mRNA with real-time reverse transcription polymerase chain reaction (RT-PCR). The mRNA expression level of PPARβ/δ was higher in the PPARβ/δ group than in the vehicle group at 6 h (Figure 1v).

### 2.2. Anti-Inflammatory Effect of PPARβ/δ

Next, we examined the anti-inflammatory effect of PPARβ/δ. We performed Naphthol AS-D chloroacetate esterase (EST) staining, nuclear factor kappa B (NF-κB) immunostaining, and kappa light polypeptide gene enhancer in the B-cell inhibitor, alpha (I-kBα) immunostaining. Furthermore, inflammatory cytokines were quantitatively examined with RT-PCR. EST-positive neutrophils were found in the corneal limbus in the early phase after injury (Figure 2a,b). On day 4, neutrophil infiltration was observed in the corneal center (Figure 2c,d). The number of neutrophils was significantly lower in the PPARβ/δ group than in the vehicle group on day 4 (Figure 2e). Inflammatory cells observed at the corneal limbus coincided with the NF-κB-positive cells. The PPARβ/δ group exhibited a smaller degree of inflammatory cell infiltration for each of the time points (Figure 3a–d). In the inflammatory cells after the alkali injury, NF-κB expression was localized in the nucleus area in the vehicle group at 6 h, while it was expressed in the cytoplasm in the PPARβ/δ group. I-κBα was strongly expressed in the cell nuclei in the PPARβ/δ group at 6 h after the alkali burn (Figure 4b). In the PPARβ/δ group, I-κBα expression was still clearly observed at day 4 after the injury (Figure 4d). The number of I-κBα-positive cells was higher in the PPARβ group as compared to that observed in the vehicle group at each of the time points (Figure 4e). Lower mRNA expression levels of interleukin-1β (IL-1β), IL-6, and NF-κB were present in the PPARβ/δ group compared to the vehicle group (Figure 5a–c). On the other hand, I-κBα mRNA was highly expressed in the PPARβ/δ group (Figure 5d). These results indicated that PPARβ/δ agonists suppressed inflammation after alkali injury.

### 2.3. Involvement of PPARβ/δ in Macrophage Polarization

To investigate macrophage polarity, we performed immunostaining with a CD68 antibody (ED-1) (Figure 6a,b) and a CD163 antibody (ED-2) (Figure 6c,d). Pan-macrophages are stained with ED-1, and M2 macrophages are stained with ED-2. The number of ED-1-positive cells was significantly lower in the PPARβ/δ group than in the vehicle group on days 1, 4, and 7 (Figure 6e). On the other hand, the number of ED-2-positive cells was significantly higher in the PPARβ/δ group than in the vehicle group on day 4 (Figure 6f). These results suggest that administration of the PPARβ/δ agonist suppresses infiltration of macrophages, while increasing M2 macrophages. In addition, the RNA expression levels of M1 and M2 macrophage selective markers and cytokines were examined by using RT-PCR analysis. Inducible nitric oxide synthase (iNOS) and tumor necrosis factor alpha (TNF-α) were measured to identify M1 macrophages, whereas arginase 1 (Arg-1) and mannose receptor (CD206) were used to identify M2 macrophages. Lower mRNA expression levels of iNOS and TNF-α were present in the PPARβ/δ group (Figure 6g, h) than in the vehicle group. Moreover, the mRNA expression levels of Arg-1 and CD206 increased in the PPARβ/δ group (Figure 6i,j). The RT-PCR results supported the immunostaining results.

### 2.4. Promotion of Neovascularization

Immunostaining for nestin and aminopeptidase P (JG12) was performed to evaluate neovascularization (Figure 7a–d). Nestin stains immature vascular endothelial cells associated with angiogenesis, whereas JG12 stains mature vascular endothelial cells. In both groups, cells that were immunopositive for nestin and aminopeptidase P were observed in the corneal limbus from day 4. Nestin-positive cells were higher on day 4, and JG12-positive cells were higher on days 1 and 4 in the PPARβ/δ group than in the vehicle group (Figure 7e,f). Lower mRNA expression levels of vascular endothelial growth factor-A (VEGF-A) were present in the PPARβ/δ group than in the vehicle group at 6 h (Figure 7g). mRNA expressions of Angiopoietin-1 (Ang-1) and Angiopoietin-2 (Ang-2) were not significantly different between the two groups (Figure 7h,i). These results suggest that PPARβ/δ agonists promote neovascularization by upregulating VEGF-A expression.

### 2.5. Effects of the PPARβ/δ Ligand on the Normal Cornea

The results shown above suggest that the PPARβ/δ agonist has a proangiogenic effect in the wound healing process. Therefore, we investigated whether administration of the PPARβ/δ agonist induces angiogenesis in the normal cornea. The PPARβ/δ agonist or vehicle was instilled onto the normal cornea twice a day, and rats were sacrificed on day 7. Histological analysis showed no obvious difference between the two groups (Figure 8).

## 3. Discussion

Alkali injury of the cornea is one of the most devastating ophthalmic conditions [12] and can cause epithelial defects, acute inflammation, neovascularization, and decreased transparency of the cornea, resulting in blindness. The purpose of this study was to demonstrate the effect of a PPARβ/δ agonist using a rat corneal alkali burn model, which is widely used to study the mechanisms of treatments, acute inflammation, and neovascularization in the injured cornea [13,14]. In the present study, we used a synthetic PPARβ/δ-specific agonist, GW501516, that has been used in research of physiological and pathophysiological functions of PPARβ/δ in conditions including obesity, diabetes, dyslipidemia, and cardiovascular disease [15,16,17]. Unfortunately, GW501516 was found highly carcinogenic [18], so its clinically use has never been practical, however, when a small volume of GW501516 is instilled as ophthalmic solutions, its carcinogenicity may not be a high risk. PPARs can be activated in ligand-dependent and ligand-independent mechanisms [19], and administration of GW501516 has been reported to increase mRNA and protein of PPARβ/δ [20,21]. Agreeing with previous studies, instilling of GW501516 increased the levels of PPARβ/δ mRNA in this study, suggesting that the expression of PPARβ/δ was ligand-dependently increased.

Although few reports have examined the anti-inflammatory effect of PPARβ/δ in the field of ophthalmology, such effects of PPARβ/δ have been reported in studies of atherosclerosis related to chronic inflammation [22,23]. PPARβ/δ activation inhibits inflammatory cytokines (such as IL-1 and IL-6) by suppressing the NF-κB pathway [24,25,26]. The results of our study are consistent with previous studies; administration of GW501516 suppressed neutrophil and macrophage infiltration and reduced inflammatory cytokines. NF-κB is a major transcription factor in cell proliferation, inflammation, immune responses, and neovascularization [27,28]. When unstimulated, it is sequestered in the cytoplasm by I-κBα, which is the suppressor protein that binds to NF-κB. I-κBα inhibits nuclear translocation of NF-κB, and as a result, transcription downstream of the NF-κB signaling pathway is restricted. Immunostaining revealed that PPARβ/δ ligand increased the expression of I-κBα. In addition, we found that there was a significant upregulation of mRNA expression of I-κBα in the PPARβ/δ group as compared to the controls. Therefore, the anti-inflammatory mechanism of the PPARβ/δ agonist may inhibit the translocation of NF-κB into the nucleus via the upregulation of I-κBα.

Macrophages exist as multiple phenotypes that can be broadly classified as M1 and M2 phenotypes [29,30]. Macrophages play different roles depending on the phenotype. M1 macrophages are pro-inflammatory, have bactericidal action, and are phagocytic. On the other hand, M2 macrophages are anti-inflammatory, promote angiogenesis, promote wound healing, and produce matrix [31]. Therefore, in addition to the degree of macrophage infiltration, the balance in macrophage phenotypes is important to determine the level of inflammation. PPARβ/δ ligands regulate macrophage polarization to the M2 phenotype in animal models of liver injury [32] and atherosclerosis [33]. In the experiments in our current study, the PPARβ/δ ligand suppressed pan-macrophage infiltration and increased M2 macrophages. The influence of PPARβ/δ on regulation of the macrophage phenotype plays a role during inflammation. However, many issues are unclear regarding macrophage polarization, and further investigation is needed.

In contrast to the anti-inflammatory effect, the PPARβ/δ agonist promoted neovascularization caused by alkali injury. Neovascularization induced by alkali injury leads to the irregular arrangement of collagen, aggravates corneal scarring and results in a loss of corneal transparency [12]. Therefore, the result of GW501516 promoting neovascularization is not desirable. PPARβ/δ-activation potently induces angiogenesis by human vascular endothelial cells in the tumor extracellular matrix in vitro and in the murine matrigel plug model in vivo [34]. Other studies have shown that PPARβ/δ ligands induce VEGF in bladder [35], breast, and prostate cancer cells [36]. In the field of ophthalmology, GW501516 was reported to promote angiogenesis in the phototherapeutic keratectomy model [37]. In the present study, as in the previous report, GW501516 promoted neovascularization pathologically and increased VEGF mRNA expression, but had no effect on Ang-1 and Ang-2 mRNA. The role of PPARβ/δ agonists in the corneal alkali burn injury was similar to that in other organs.

Based on this result, we examined whether the angiogenesis-promoting effect of the PPARβ/δ ligand was present in normal rat corneas. After 7 days of administration of GW501516 twice a day, no apparent change was seen in macroscopic or pathological images.

Promotion of VEGF expression by PPARβ/δ appears to require inflammation. Thus, further investigation is needed to assess the two conflicting actions (anti-inflammatory and pro-angiogenic) prior to the use of PPARβ/δ in ophthalmology. 

## 4. Materials and Methods 

### 4.1. Animals and Ethics Statement

Eight-week-old male Wistar rats (Sankyo Laboratory Service, Tokyo, Japan) were used for all experiments in this study. All animal experiments were conducted in compliance with the Experimental Animal Ethics Review Committee of Nippon Medical School (approval number: 2019-005, 1 April 2019) in Tokyo, Japan, and all procedures conformed to the Association for Research in Vision and Ophthalmic and Visual Research. 

### 4.2. Experimental Procedures

Under general isoflurane anesthesia, a circular filter paper 3.2 mm in diameter was immersed in 1 N NaOH and placed on the central cornea of each rat for 1 min to create a corneal alkali burn. After alkali exposure, the ocular surface was washed with 40 mL physiological saline. PPARβ/δ agonist ophthalmic solution or vehicle solution was administered twice a day to the alkali-burned cornea. The vehicle solution was filter purified 100mL NaCl-based PBS (0.01 M; pH 7.4), which was prepared with 232 g disodium hydrogen-phosphate 12-water, 23.7 g sodium dihydrogen phosphate dihydrate, 4000 mL distilled water, and 0.1 mL polyoxyethylene sorbitan monooleate (Wako Pure Chemical Industries, Osaka, Japan). The PPARβ/δ agonist ophthalmic solution was prepared by adding 10 mg GW501516 to 20 mL vehicle solution. At each endpoint (6 h, 1 day, 4 days, and 7 days after alkali burn), rats were sacrificed by exsanguination under isoflurane anesthesia. Pathological and molecular biological evaluations were performed on the enucleated eyes.

### 4.3. Histological and Immunohistochemical Analyses

The excised eyes were fixed with 10% buffered formalin and embedded in paraffin. For histopathological examination, HE staining was performed on deparaffinized tissue. EST staining was performed to detect infiltrating neutrophils. We used monoclonal mouse anti-rat ED-1 (BMA, Nagoya, Japan), monoclonal mouse anti-rat ED-2 (BMA), polyclonal rabbit anti-NF-kB/P65 (Santa Cruz Biotechnology, Dallas, TX, USA) and monoclonal mouse anti- I-kBα (Santa Cruz Biotechnology) as primary antibodies for immunohistochemical analysis. Angiogenesis was evaluated using monoclonal mouse anti-nestin (Nestin; Merck Millipore, Darmstadt, Germany [38]) and monoclonal mouse anti-JG12 (Thermo Fisher Scientific, Waltham, MA, USA [39]) antibodies. Three places in the central cornea and two places in the limbus were observed at a magnification of 400×. The number of each type of positively stained cells was counted, and the average number was calculated.

### 4.4. Real-Time RT-PCR

Using real-time RT-PCR (Thermo Fisher Scientific), we investigated the mRNA expression of PPARβ/δ, iNOS, TNF-α, Arg-1, CD206, IL-1β, IL-6, NF-κB, I-κBα, VEGF-A, Ang-1, and Ang-2. Total RNA was extracted from the enucleated corneas using a RNeasy^®^ FFPE Kit (Qiagen, Hilden, Germany). We treated total RNA with DNase I (RNase-free DNase Set; Qiagen) to remove contaminating DNA. All samples showed an OD 260/280 nm ratio >1.8. The High Capacity cDNA Reverse Transcription Kit (Thermo Fisher Scientific) was used to reverse transcribe first-strand cDNA from total RNA (100 ng) in a volume of 20 μL. The cDNA (5 μL) was used for each PCR (total reaction volume, 20 μL). According to the manufacturer′s instructions, optimized primers and probes were used for each target gene. The primers used in this experiment are described below (Table 1). PCR (2 min at 50 °C, 10 min at 95 °C, and 45 cycles of denaturation at 95 °C for 15 s and annealing at 60 °C for 60 s) was conducted on a QuantStudio™ 3 Real-Time PCR System (Thermo Fisher Scientific) according to fluorescent TaqMan methodology. The mRNA levels were determined in duplicate, and the expression of each mRNA was normalized to that of β-actin. We calculated the relative expression levels using the 2^−ΔΔCT^ method [40].

### 4.5. Statistical Analyses 

All results are expressed as the mean ± standard deviation. Statistical analyses were performed using unpaired Student’s t-test using Excel analytical software (Excel, Microsoft, Redmond, WA, USA). Values of *p* < 0.05 were considered statistically significant.

## 5. Conclusions

In summary, the PPARβ/δ agonist showed anti-inflammatory effects and promoted neovascularization in a rat alkali burn model. PPARβ/δ agonist administration did not cause neovascularization or inflammation in the normal cornea. 

## Figures and Tables

**Figure 1 ijms-21-05296-f001:**
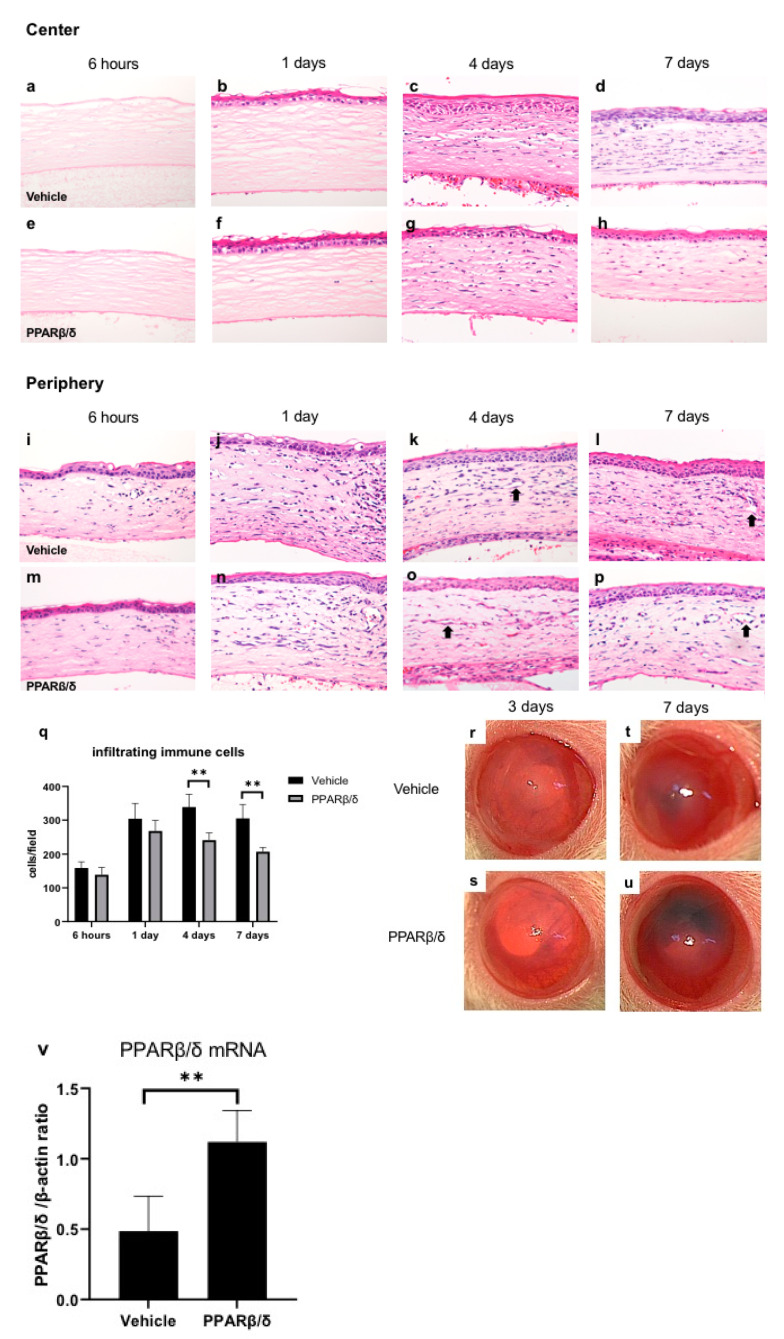
Wound healing process after alkali burn and expression of PPARβ/δ mRNA. The time-dependent changes in the vehicle group (**a–d**: central region, **i–l**: peripheral region) and PPARβ/δ group (**e–h**: central region, **m–p**: peripheral region). Bar, 50 μm. After the alkali burn in both groups, inflammatory cells infiltrated from the corneal limbus into the central area by day 7. On day 4 and day 7, luminal structures (black arrows) due to neovascularization were observed. The number of infiltrating cells was lower in the PPARβ/δ group than in the vehicle group (**q**). The macroscopic photographs of anterior segment on day 3 (**r**: vehicle group, **s**: PPARβ/δ group) and day 7 (**t**: vehicle group, **u**: PPARβ/δ group). In the PPARβ/δ group, corneal transparency was better than the vehicle group on day 3, but on day 7, anterior chamber bleeding was deteriorated and transparency was lost. Regarding mRNA expression levels of PPARβ/δ, administration of the PPARβ/δ ligand significantly upregulated PPARβ/δ expression in the cornea at 6 h (**v**). ** *p* < 0.01.

**Figure 2 ijms-21-05296-f002:**
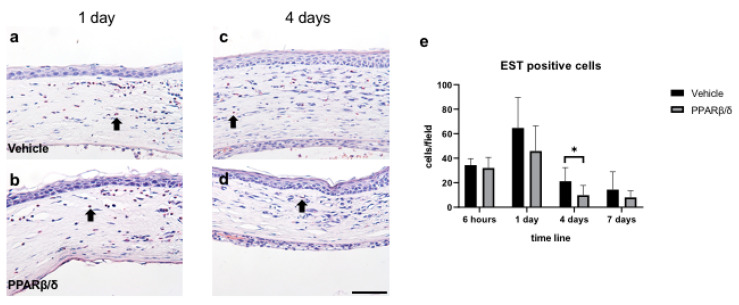
EST staining. EST staining was performed to examine neutrophil infiltration. Vehicle group (**a**,**c**) and PPARβ/δ group (**b**,**d**) in the corneal limbus on day 1 and day 4 after injury (black arrows: EST-positive cells). Bar, 50 μm. Fewer EST-positive neutrophils were present in the PPARβ/δ group than in the vehicle group (**e**). Data are presented as mean ± SD (*n* = 8) * *p* < 0.05.

**Figure 3 ijms-21-05296-f003:**
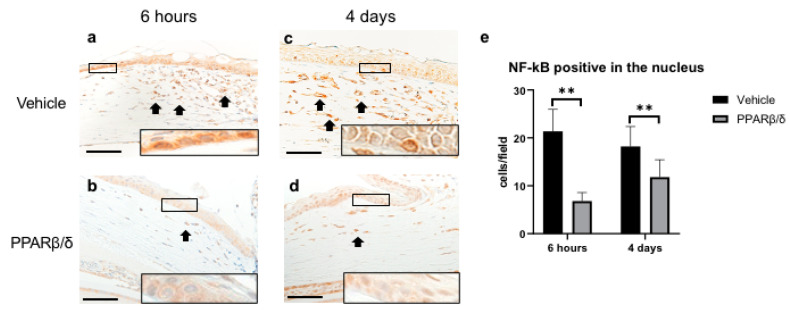
Immunostaining of NF-κB and quantitative evaluation analyzed by cell counting on the corneal periphery in each group. (**a**,**b**) Representative sections at 6 h after the injury are shown. As compared to the PPARβ/δ group, the vehicle group exhibited strong staining in the nucleus of inflammatory cells (see boxed area). (**c**,**d**) Representative sections at day 4 after injury are shown. Higher magnification pictures of the boxed area are also shown. NF-κB-positive inflammatory cells (black arrows) were observed in the corneal limbs. Bar, 50 μm. The number of cells stained in the nucleus was significantly lower in the PPARβ/δ group versus the vehicle group at each of the time points (**e**). Data are presented as mean ± SD (*n* = 8) ** *p* < 0.01.

**Figure 4 ijms-21-05296-f004:**
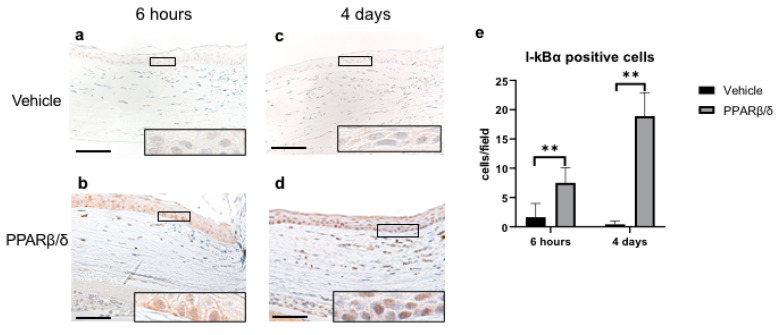
Immunohistochemical analysis of I-κBα and quantitative evaluation determined by cell counting on the corneal periphery in each group. Representative sections at 6 h after injury are shown (**a**,**b**). PPARβ/δ group exhibited strong expression of I-kBα as compared to the vehicle group (see boxed area). Representative sections at day 4 after the injury are shown. I-κBα expression was only found in the PPARβ/δ group (**c**,**d**). Higher magnification pictures of the boxed area are also shown. Bar, 50 μm. There was a significantly higher number of I-κBα-positive cells in the PPARβ group versus the vehicle group at each of the time points (**e**). Data are presented as mean ± SD (*n* = 8) ** *p* < 0.01.

**Figure 5 ijms-21-05296-f005:**
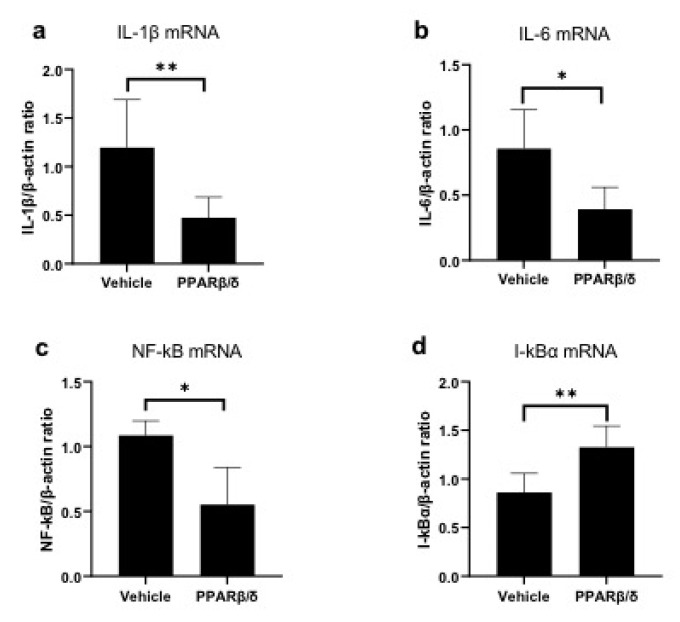
Expression of mRNA for inflammatory cytokines. Quantification of mRNA levels of IL-1β (**a**), IL-6 (**b**), and NF-κB (**d**) at 6 h. Expression of inflammatory cytokines and NF-κB was significantly lower in the PPARβ/δ group than in the vehicle group (**a**–**c**). The mRNA expression levels of I-κBα were higher in the PPARβ/δ group as compared to that observed for the vehicle group (**d**). Data are presented as mean ± SD (*n* = 8) ** *p* < 0.01 or * *p* < 0.05.

**Figure 6 ijms-21-05296-f006:**
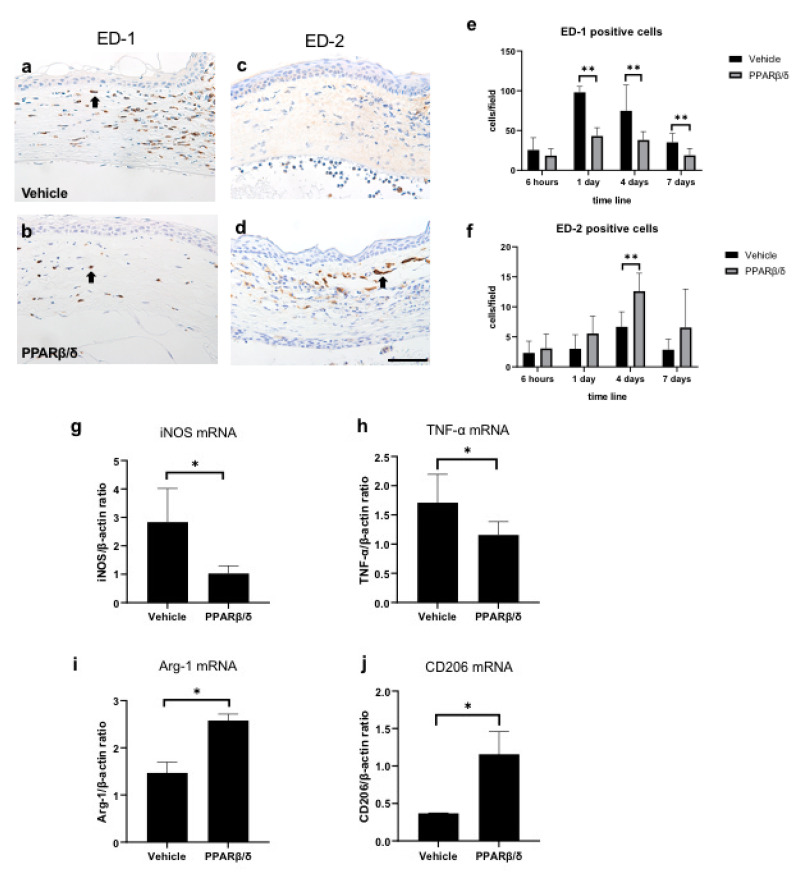
Immunostaining for ED-1 and ED-2. Pan-macrophages are stained with ED-1, and M2 macrophages are stained with ED-2. ED-1 immunostaining (**a**: vehicle group, **b**: PPARβ/δ group) in the corneal limbus on day 1. ED-2 immunostaining (**c**: vehicle group, **d**: PPARβ/δ group) in the corneal limbus on day 4. Black arrows: immunostained cells. Bar, 50 μm. ED-1-positive cells were lower on days 1, 4, and 7. On the other hand, ED-2-positive cells were higher on day 4 in the PPARβ/δ group versus the vehicle group (**e**, **f**). The mRNA levels of the indicated M1 (iNOS and TNF-α) (**g**, **h**) and M2 (Arg-1, and CD206) (**I**, **j**) at 6 h. The M1 markers were lower and the M2 markers were higher in the PPARβ/δ group as compared to the vehicle group. Data are presented as mean ± SD (*n* = 8) ** *p* < 0.01 or * *p* < 0.05

**Figure 7 ijms-21-05296-f007:**
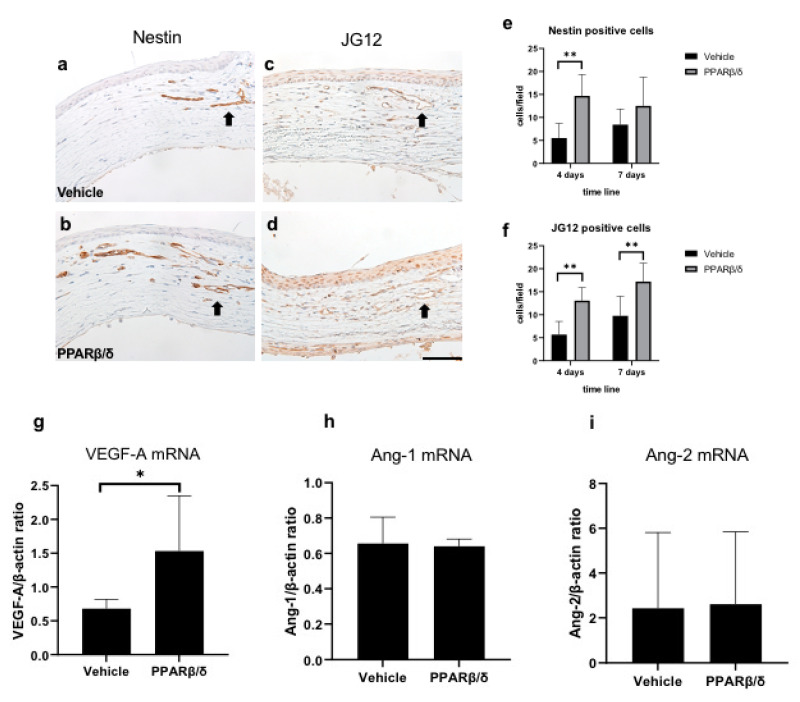
Evaluation of neovascularization. Nestin immunostaining in the corneal limbus on day 4 (**a**: vehicle group, **b**: PPARβ/δ group). JG12 immunostaining in the corneal limbus on day 7 (**c**: vehicle group, **d**: PPARβ/δ group). Black arrows: immunopositive cells. Bar, 50 μm. Nestin-positive endothelial cells, and JG12-positive capillary lumens were observed from day 4. The number of nestin-positive cells was higher on day 4 in the PPARβ/δ group compared to the vehicle group (**e**). The number of JG12-positive cells was higher on days 4 and 7 in the PPARβ/δ group compared to the vehicle group (**f**). Among cytokines related to angiogenesis, only VEGF-A showed a significant difference between the PPARβ/δ group and vehicle group (**g–i**). Data are presented as mean ± SD (*n* = 8) ** *p* < 0.01 or * *p* < 0.05.

**Figure 8 ijms-21-05296-f008:**
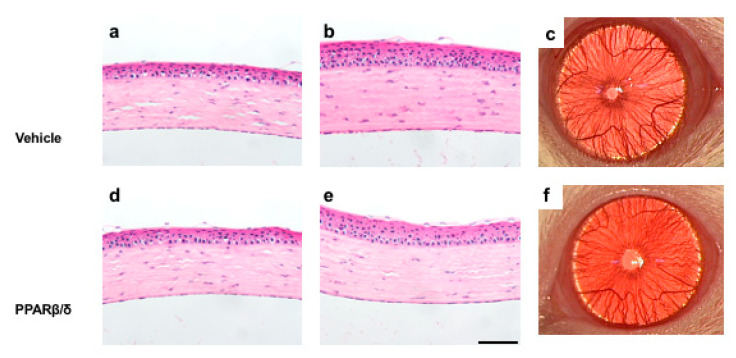
Effect of the PPARβ/δ agonist on the normal cornea at 7 days. HE staining of the vehicle group (**a**: central region, **b**: peripheral region) and PPARβ/δ group (**d**: central region, **e**: peripheral region). Bar, 50 μm. No obvious difference in the HE staining images or macroscopic images (**c**, **f**) was apparent between the two groups.

**Table 1 ijms-21-05296-t001:** Primer pairs used in this study.

Gene	Forward Primer Sequence (5′-3′)	Reverse Primer Sequence (5′-3′)
β-actin	GCAGGAGTACGATGAGTCCG	ACGCAGCTCAGTAACAGTCC
iNOS	TCACCTTCGAGGGCAGCCGA	CAGACGCCATGGTGCAGGGG
TNF-α	AAATGGGCTCCCTCTCATCAGTTC	TCTGCTTGGTGGTTTGCTACGAC
Arg-1	ATTCACCCCGGCTACGGGCA	AGGAGCAGCGTTGGCCTGGT
CD206	GACGGACGAGGAGTTCATTATAC	GTTGGAGAGATAGGCACAGAAG
IL-1β	TACCTATGTCTTGCCCGTGGAG	ATCATCCCACGAGTCACAGAGG
IL-6	GTCAACTCCATCTGCCCTTCAG	GGCAGTGGCTGTCAACAACAT
NF-κB	TGGACGATCTGTTTCCCCTC	TCGCACTTGTAACGGAAACG
I-κBα	TGACCATGGAAGTGATTGGTCAG	GATCACAGCCAAGTGGAGTGGA
VEGF-A	GCAGCGACAAGGCAGACTAT	GCAACCTCTCCAAACCGTTG
PPARβ/δ	GCCGCCCTACAACGAGATCA	CCACCAGCAGTCCGTCTTTGT

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
