# Peer review of "Peroxisome Proliferator-Activated Receptor Beta/Delta Agonist Suppresses Inflammation and Promotes Neovascularization"

_ijms, 2020, doi:10.3390/ijms21155296_

Round 1
Reviewer 1 Report
Based on the previous effects of PPARβ/δ agonist against inflammatory cytokines of atherosclerosis authors tried to assess the effect of this agonist on the cornea wound in this manuscript. Authors applied peroxisome proliferator-activated receptor (PPAR)β/δ agonist, GW501516, on a rat corneal alkali burn model. The result showed that GW501516 suppressed the infiltration of neutrophils and macrophage and NF-kB dependent inflammation while it promoted neovascularization in the corneal wound healing process.
- In the result section, authors did not specify the name of ppar agonist. For example, they simply wrote “ Hematoxylin-eosin (HE) staining was conducted to compare the corneal wound healing process between the vehicle group and the PPARβ/δ group (the first sentence of the result). They should specify (PPAR)β/δ agonist, GW501516 instead of just mentioning (PPAR)β/δ group.
- They should have mRNA levels of M1 and M2 makers to support IHC data.
- In figure 1, the number of infiltrating immune cells should be quantified
Author Response
We are grateful for the opportunity to revise and resubmit our paper (ijms-850390) entitled, “Peroxisome proliferator-activated receptor beta agonist suppresses inflammation and promotes neovascularization”, and would like to thank you for helpful comments.
Attached to this letter is a revised version of our manuscript that shows the tracked changes. In addition, we have separately listed our point-by-point responses to the comments of the reviewer in the text that follows.
We would like to thank you for constructive comments. After incorporating the suggested changes, this has greatly improved our paper, thereby making this a much stronger manuscript. Please convey our thanks to you for helpful input.
Based on the previous effects of PPARβ/δ agonist against inflammatory cytokines of atherosclerosis authors tried to assess the effect of this agonist on the cornea wound in this manuscript. Authors applied peroxisome proliferator-activated receptor (PPAR)β/δ agonist, GW501516, on a rat corneal alkali burn model. The result showed that GW501516 suppressed the infiltration of neutrophils and macrophage and NF-kB dependent inflammation while it promoted neovascularization in the corneal wound healing process.
→Thank you for your kind comments.
1.In the result section, authors did not specify the name of ppar agonist. For example, they simply wrote “ Hematoxylin-eosin (HE) staining was conducted to compare the corneal wound healing process between the vehicle group and the PPARβ/δ group (the first sentence of the result). They should specify (PPAR)β/δ agonist, GW501516 instead of just mentioning (PPAR)β/δ group.
→We added to the introduction (line 45) and the beginning of the results (line 51) that GW501516 was used as a PPARβ/δ agonist.
2.They should have mRNA levels of M1 and M2 makers to support IHC data.
→We added the research of mRNA levels of M1 and M2 makers to support IHC data (line 139-146 and Figure 6).
3.In figure 1, the number of infiltrating immune cells should be quantified.
→We counted the number of infiltrating immune cells and the result was added to figure 1.
Once again, thank you for all of the helpful suggestions for further revising our manuscript. If you have any further questions or require other information, please feel free to contact us.
Sincerely,
Yutaro Tobita, M.D
Department of Ophthalmology, Nippon Medical School
1-1-5 Sendagi, Bunkyo-ku, Tokyo 113-8603, Japan
E-mail: y-tobita@nms.ac.jp
Reviewer 2 Report
In the manuscript by Tobita et al, the authors to investigate the therapeutic effects of Peroxisome proliferator-activated receptor beta agonist in corneal alkali injury. The research approach to the questions were appropriate and resulted in quality data. They showed significant anti-inflammatory effect by the ligand but the ligand worsening the neovascularization. There is merit to report the effect of topical use of PPARβ/δ agonist in corneal alkali injury. Macroscopic images showing the effect of GW501516 for the alkali injured eye are required to assess the overall effect.
GW501516 should be properly introduced in the introduction and the previous use and potential side effects should be addressed in the discussion. The significance of neovascularization during cornea alkali injury should be properly discussed. In previous reported studies of GW01516, are there any effects of GW01516 in vascularization observed?
Minor points:
The meaning of Figure 1q should be discussed. Why the treatment of agonist increases the expression of receptor? How widely such phenomena are observed? Biological meaning?
In methods line 226: Is the filter paper placed on the cornea for 1 min or immersed in NaOH for 1 min?
in methods line 229: In biology, saline means NaCl in water. The following should be the proper expression.
“The vehicle solution was filter purified 100 ml phosphate-buffered saline (0.01 229 M; pH 7.4), “
in methods line 243: ‘ ) ’ was missed.
Author Response
We are grateful for the opportunity to revise and resubmit our paper (ijms-850390) entitled, “Peroxisome proliferator-activated receptor beta agonist suppresses inflammation and promotes neovascularization”, and would like to thank you for helpful comments.
Attached to this letter is a revised version of our manuscript that shows the tracked changes. In addition, we have separately listed our point-by-point responses to the comments of the reviewer in the text that follows.
We would like to thank you for constructive comments. After incorporating the suggested changes, this has greatly improved our paper, thereby making this a much stronger manuscript. Please convey our thanks to you for helpful input.
In the manuscript by Tobita et al, the authors to investigate the therapeutic effects of Peroxisome proliferator-activated receptor beta agonist in corneal alkali injury. The research approach to the questions were appropriate and resulted in quality data. They showed significant anti-inflammatory effect by the ligand but the ligand worsening the neovascularization. There is merit to report the effect of topical use of PPARβ/δ agonist in corneal alkali injury.
→Thank you for your kind comments.
Macroscopic images showing the effect of GW501516 for the alkali injured eye are required to assess the overall effect.
→We added macroscopic images (figure 1) and comments (line 59-64).
GW501516 should be properly introduced in the introduction and the previous use and potential side effects should be addressed in the discussion.
→We added the sentences that GW501516 was used as a PPARβ/δ agonist in the introduction (line 45), and the sentences regarding the previous use and potential side effects in the discussion (line 196-201).
The significance of neovascularization during cornea alkali injury should be properly discussed. In previous reported studies of GW501516, are there any effects of GW501516 in vascularization observed?
→According to your comment, we added the sentences regarding the significance of neovascularization during cornea alkali injury in the discussion (232-234), and the previous report of GW501516 and vascularization (237-239).
Minor points:
The meaning of Figure 1q should be discussed. Why the treatment of agonist increases the expression of receptor? How widely such phenomena are observed? Biological meaning?
→We added the sentences about the meaning of Figure 1q to the discussion (201-205).
In methods line 226: Is the filter paper placed on the cornea for 1 min or immersed in NaOH for 1 min?
→The filter paper placed on the cornea for 1 min. We moved the term "for 1 minute" to another position (line 258).
in methods line 229: In biology, saline means NaCl in water. The following should be the proper expression.
“The vehicle solution was filter purified 100 ml phosphate-buffered saline (0.01 229 M; pH 7.4), “
→In line with your comment, we revised (line 261).
in methods line 243: ‘ ) ’ was missed.
→In line with your comment, we revised (line 274).
Once again, thank you for all of the helpful suggestions for further revising our manuscript. If you have any further questions or require other information, please feel free to contact us.
Sincerely,
Yutaro Tobita, M.D
Department of Ophthalmology, Nippon Medical School
1-1-5 Sendagi, Bunkyo-ku, Tokyo 113-8603, Japan
E-mail: y-tobita@nms.ac.jp
Round 2
Reviewer 1 Report
The authors answered all the questions requested.
There is no further comment.
Reviewer 2 Report
The manuscript was revised properly.